# Assessing Project Proposals Based on National and Global Tiger Action Plans: Lessons from the Integrated Tiger Habitat Conservation Programme (ITHCP)

**Johan Diepstraten [1,\*], Mitali Sharma [2], Mohammad Khalid Sayeed Pasha [3,4] and Sugoto Roy [5]**

1   Animal Behaviour and Cognition, Department of Biology, Faculty of Science, Utrecht University, Padualaan 8, 3584 CH Utrecht, The Netherlands
2   Independent Researcher, Singapore 238275, Singapore
3   WWF Singapore, 354 Tanglin Road, Tanglin Block, Tanglin International Centre, Singapore 247672, Singapore
4   Science and Strategy Group, IUCN Asia Regional Office, 63 Sukhumvit Soi 39, Bangkok 10110, Thailand
5   IUCN SSC Cat Specialist Group, c/o Foundation KORA, Talgut-Zentrum 5, CH-3063 Ittigen, Switzerland
\*   Correspondence: johandiepstraten@gmail.com

**Abstract:** Tigers play a crucial role in maintaining healthy ecosystems. Unfortunately, tigers are threatened by poaching, human–wildlife conflict, habitat loss, and more. In response to these threats, the conservation community pledged to double the worldwide wild tiger population by 2022 (known as TX2) at the "Tiger Summit" in St. Petersburg in 2010, and to track the progress of Tiger Range Countries. Between 2010 and 2022, the Global Tiger Recovery Programme was implemented. To accomplish this TX2 goal, each Tiger Range Country developed a National Tiger Action Plan (NTAP). The Integrated Tiger Habitat Conservation Programme (ITHCP) is a grant-making mechanism that focusses on a subsection of the Global Tiger Recovery Programme. It had twelve projects in six Tiger Range Countries during Phase 1 of the program. Evaluating the proposals of these projects is crucial for resource allocation. In this study, we assessed project proposals by evaluating how the proposed activities of all twelve ITHCP projects addressed their corresponding NTAPs, by comparing the plans against the proposals. A further comparison was undertaken using the Conservation Assured | Tiger Standards Lite, a site-based tiger conservation accreditation system. Overall, this study shows the importance of both global and national action plans and how comparing project activities with NTAP requirements can help address resource allocation needs to fill gaps in management. We conclude that projects should be designed to closely align with national action plans, best practice standards, and the activities of other projects in their landscape to maximize conservation outputs and impact. However, projects on their own are not enough to satisfy whole NTAPs.

**Keywords:** tiger conservation; conservation management; project design; action plan; best practices

## 1. Introduction

As the world is beginning to develop frameworks for the post-2020 conservation era, we need to look at conservation successes and failures and gaps in our approach to date. The conservation crisis is epitomized by problems faced by large apex predators such as tigers. Besides their cultural value, these predators also play a crucial role in keeping ecosystems healthy and diverse [1,2]. Due to their position at the top of the food chain, they help maintain balance between herbivore populations and the vegetation they feed upon [3]. Tigers themselves are also considered to be an umbrella species [4]. Ranging throughout ecosystems that support some of the world's richest biodiversity, tiger habitats provide niches for many globally threatened animal and plant species [1,5]. These healthy ecosystems provide multiple ecosystem services to the planet, such as carbon sequestration and regulation of watersheds that supply up to 830 million people with access to water sources [1,6]. Tiger landscapes also offer economic benefits by supplying local communities

with sustainable natural resources and creating opportunities for local people to take part in alternative income generation activities such as ecotourism—the most profitable sector in the tourism industry [1,7]. However, the number of wild tigers has declined rapidly, with numbers falling from 100,000 at the start of the last century to below 3500 in 2010 [8]. Where they once ranged widely across Asia, tiger populations now occupy less than 6% of their historical range [9]. This rapid decline can be attributed to several threats faced by tigers, such as poaching of tigers and their prey, fragmentation and loss of habitat, human–wildlife conflict and climate change [2,10–13]. The same problem is seen for other large predators such as jaguars, lions and leopards across their range.

In 2008, the World Bank hosted the launch of an alliance between international organizations, civil society organizations, and the governments of all Tiger Range Countries called the Global Tiger Initiative [14]. This collaboration resulted in the Global Tiger Recovery Programme: a structural framework that is based on the conservation needs and threat assessments for tigers to form a plan to double wild tiger populations by 2022—the Year of the Tiger [15]. The plan aimed to be accomplished through (i) the effective management, preservation, protection, and enhancement of tiger habitats, (ii) the eradication of tiger poaching, smuggling, and illegal trade, (iii) cooperation in transboundary landscape management to combat illegal trade, (iv) engaging with local communities, (v) increasing management effectiveness, and (vi) restoring tigers to their former range [15]. These actions focused on 76 potential tiger habitats called Tiger Conservation Landscapes [9]. Since not all Tiger Range Countries have the same potential of doubling their tiger populations, the exact actions to conform to the Global Tiger Recovery Programme differed per country [15]. Therefore, the building blocks of the Global Tiger Recovery Programme are the National Tiger Action Plans (NTAPs) of each Tiger Range Country [15]. In these NTAPs, each country defines actions that are specific to their national tiger status and context to increase local tiger populations and contribute to the Global Tiger Recovery Programme. Where there is a strong government focus on tiger conservation, government agencies partner with conservation projects and have continuous input into project design from inception to implementation. In countries where governments do not have a strong tiger-focused conservation agency input, it is often the NGOs that feed into NTAPs. As a result, NTAPs are meant to be a locally appropriate tool that improves the planning, implementation, management, monitoring, and evaluation of projects.

One international organization involved in tiger conservation is the International Union for Conservation of Nature (IUCN). In 2014, with funding from the German Government (Ministry for German Cooperation) and the German Development Bank KfW, they initiated a grant-making mechanism called the Integrated Tiger Habitat Conservation Programme (ITHCP) [16]. This program is based on three pillars, which are a subset of the Global Tiger Recovery Programme objectives, namely, (i) the appropriate management of tiger habitats, including buffer zones and corridors, (ii) the protection of tiger and prey populations from poaching, and (iii) improvements in the livelihoods of human populations living in and around tiger habitats [16]. The program consists of 12 large-scale projects in Tiger Conservation Landscapes located in Bangladesh, Bhutan, India, Indonesia, Myanmar, and Nepal (Figure 1) [16]. Most (seven) of the ITHCP projects came to an end in 2021, but five of them will continue into Phase 2 of the ITCHP. Therefore, it is important to address resource allocation needs. To do this, it is crucial to monitor and evaluate the design of tiger conservation projects. Many studies have undertaken such assessments [17–20]. However, this study explores how project proposals met the requirements of the ITHCP and aligned themselves to the NTAPs of their respective countries. To achieve this, firstly, the overlap between the NTAPs and the mentioned subset of Global Tiger Recovery Program objectives was analyzed. Thereafter, the proposals for each ITHCP project were compared to their corresponding NTAPs to see to what extent proposed activities meet the national requirements that correspond to the ITHCP pillars. Subsequently, unaddressed action points are identified. We used project proposals because project results take a substantial amount of

time to be released, and there are many other contributing factors to successes and failures, including other projects.

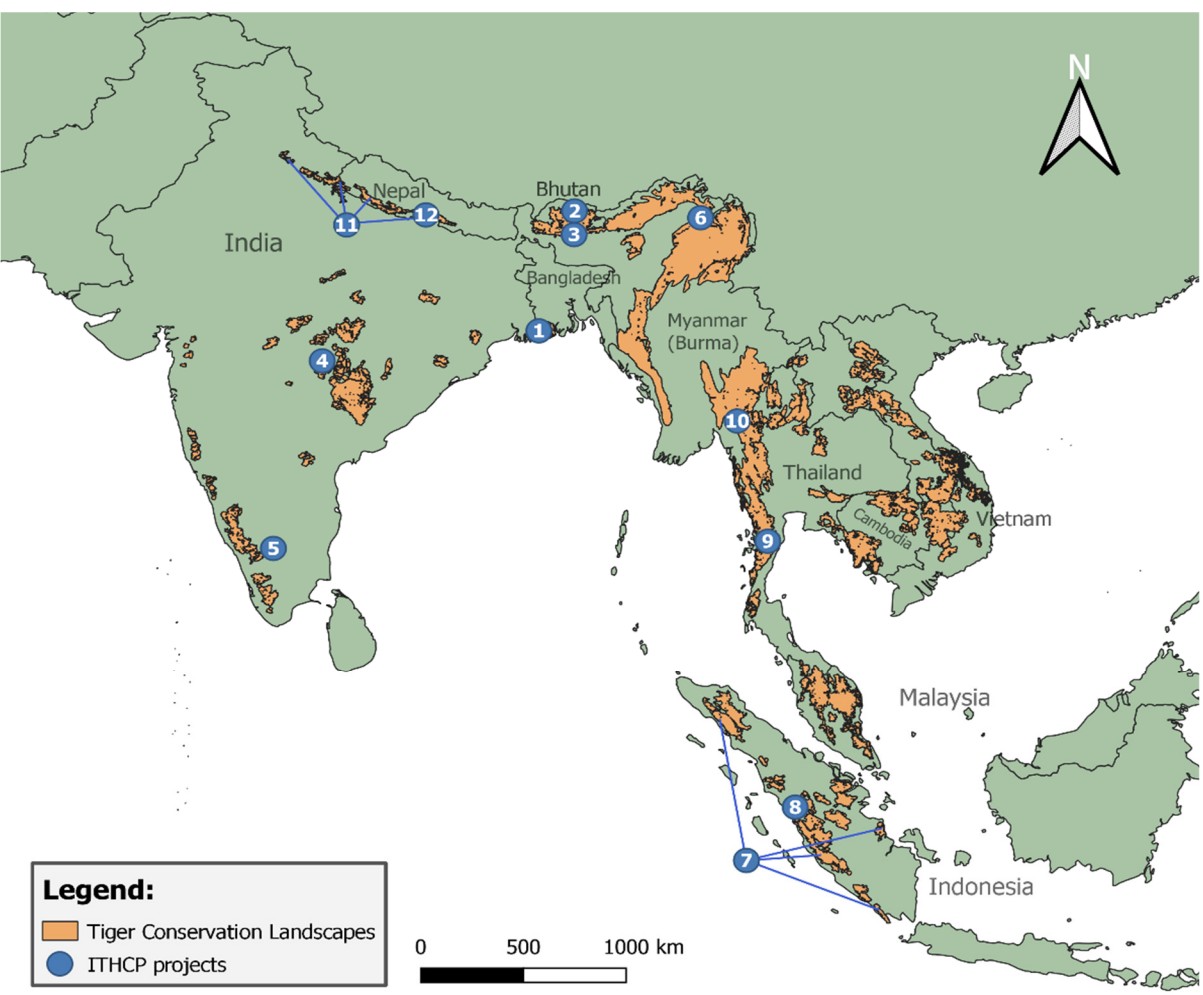

**Figure 1.** The 12 Integrated Tiger Habitat Conservation Programme (ITHCP) projects across Tiger Conservation Landscapes in 6 countries: Bangladesh, Bhutan, India, Indonesia, Myanmar, and Nepal. Projects 1, 6, 11, and 12 are transboundary projects [21].

Another part of successful resource allocation is evaluating the effectiveness of tiger conservation efforts and providing a benchmark against which progress can be measured. For this, the Conservation Assured | Tiger Standards (CA | TS) accreditation system was designed [22]. CA | TS presents a distillation of best practices and a roadmap to effective management that covers all aspects of the Global Tiger Recovery Programme [22,23]. Sites within Tiger Conservation Landscapes are assessed against these standards and, if the standards are met, accredited against CA | TS and approved. However, since CA | TS is a relatively new accreditation system, not all sites have been assessed by these standards. To expedite the process, "CA | TS Lite" was initiated [24]. This was a rapid survey conducted in 112 sites to assess the progress of conservation efforts based on 40 questions that represent the CA | TS standards. The surveys were conducted by site managers and reviewed by independent experts to ensure data validity and information verification. The answers to these questions provide a quick overview of the current progress of the Tiger Conservation Landscapes. Furthermore, they help provide insights into the challenges faced by Tiger Range Countries and create a baseline for the future implementation of CA | TS [24]. Therefore, CA | TS Lite is a useful tool to assess the current status of tiger conservation in sites where projects operate, as it is used for in this study.

Thus, to follow up on our comparison between project proposals and NTAPs, a further comparison is undertaken to see whether the unaddressed points were also identified in the CA | TS Lite assessments. This way, it is possible to both uncover whether unaddressed action points pose a problem in project design, and whether CA | TS is an appropriate diagnostic tool. Eventually, the aim of this study is to see whether it is possible to find gaps in resource needs by comparing project proposals to their corresponding national plans and project processes to best practice standards.

## 2. Materials and Methods

Since our assessment was based on the three ITHCP pillars, the pillars were broken down into the following ITHCP categories:

1. The appropriate management of tiger habitats, including buffer zones and corridors:
   - Implementation of management techniques;
   - Investment in adequate staff;
   - Investment in adequate facilities.

2. The protection of tiger and prey populations from poaching:
   - Efforts to counteract poaching;
   - Activities to monitor tiger and prey populations.

3. The engagement with human populations living in and around tiger habitats:
   - Efforts to reduce human–wildlife conflict;
   - Efforts to improve local livelihoods;
   - Efforts to increase local acceptance of conservation activities;
   - Activities to develop ecotourism.

The most recent version of each NTAP was used to identify action points that align with the three ITHCP pillars. First, to obtain a clear view of the content of the NTAPs, a summary table was made to display the publication date, the implementation period and the aspects of the ITHCP categories that were addressed in each NTAP. Thereafter, project proposals ($n$ = 12) were analyzed to see how many of the identified action points of their respective NTAPs were addressed (Supplementary Material Tables S1–S6). Action points in project proposals were generally more detailed than those stated in the NTAPs. Therefore, each action point in the NTAP that was addressed in the corresponding project proposal was scored with "1". Unaddressed action points were scored "0". Subsequently, percentages for each pillar and an overall percentage were calculated to express the similarities between project proposals and the action points identified in the NTAPs. Transboundary projects ($n$ = 4) were compared to the NTAPs of all countries in which they are located. Therefore, transboundary projects received two percentage values for each ITHCP pillar. Note that these projects would have all activities under broader categories in both countries. However, some activities are less important on one side of the border.

In addition, the CA | TS Lite assessment of each project's conservation site was analyzed for all ITHCP pillars to see whether they were sufficiently addressed. In the CA | TS Lite assessment, each of the 40 CA | TS Lite indicators is assigned a value ranging between 1 and 0:

- 1 = Importance of standard is recognized and action is implemented;
- 0.75 = Importance of standard is recognized and action is initiated;
- 0.5 = Importance of standard is recognized and action is in planning progress;
- 0.25 = Importance of standard is recognized but no action is initiated;
- 0 = Importance of standard is not recognized and action is absent.

For every ITHCP pillar, the appropriate corresponding CA | TS Lite standards were selected (Supplementary Material Table S7). The mean value was calculated for every pillar. Categories with a mean value of <0.5 were considered to be insufficiently addressed. In many cases, ITHCP projects cover numerous reserves, causing multiple CA | TS Lite

assessments to be available for the project. In these instances, values of assessments were grouped to calculate one average value per ITHCP category.

Finally, changes in tiger population size were assessed for each project by data from camera trapping and population estimates performed by the projects themselves.

## 3. Results

NTAPs do not fully cover all aspects of the ITHCP's subset of Global Tiger Recovery Programme objectives. Different countries focus on different aspects (Table 1).

**Table 1.** Overview of what focal areas of the Global Tiger Recovery Programme and ITHCP are addressed in the NTAPs for each ITHCP category. Shaded cells with bold text indicate the different clusters of focal areas. Underneath the shaded rows, the clusters are broken down in more specified focal areas. All focal areas marked with an "X" are addressed [13,25–29].

| | Bangladesh | Bhutan | India | Indonesia | Myanmar | Nepal |
|---|---|---|---|---|---|---|
| Year of publication | 2018 | 2018 | 2011 | 2007 | 2003 | 2016 |
| Implementation period | 2018–2027 | 2018–2023 | 2011–2022 | 2007–2017 | 2003–2007 | 2016–2020 |
| **Management implementation** | **X** | **X** | **X** | **X** | **X** | **X** |
| Infrastructure management | - | X | X | - | - | - |
| Core habitat management | X | X | - | X | X | X |
| Buffer zone management | X | - | - | X | X | X |
| Corridor management | - | - | X | X | X | X |
| Water management | X | X | X | - | - | X |
| Adaptive management | X | - | - | X | - | X |
| Communication | - | X | - | - | - | - |
| **Investment in staff** | **X** | **X** | **X** | **X** | **X** | **X** |
| Training | X | X | X | X | X | X |
| Welfare | X | - | X | - | X | - |
| **Investment in facilities** | **-** | **X** | **X** | **-** | **X** | **X** |
| Anti-poaching equipment | - | X | X | - | X | X |
| Patrolling equipment | - | X | X | - | - | X |
| Communication equipment | - | X | X | - | - | - |
| Anti-human–wildlife conflict equipment | - | - | X | - | - | - |
| **Anti-poaching** | **X** | **X** | **X** | **X** | **X** | **X** |
| Anti-poaching squad | X | X | X | X | X | X |
| Patrol system | X | X | - | - | - | X |
| Penalties for wildlife offenders | X | - | - | - | X | - |
| Informants | X | X | X | X | - | X |
| **Monitoring** | **X** | **X** | **X** | **X** | **X** | **X** |
| Tiger population | X | X | X | X | X | X |
| Prey population | X | X | X | X | X | X |
| **Reduce human–wildlife conflict** | **X** | **X** | **X** | **X** | **X** | **X** |
| Mitigation activities | X | X | X | X | X | X |
| Monitor human–wildlife conflict | X | X | - | - | - | X |
| **Improve local livelihoods** | **X** | **X** | **X** | **-** | **-** | **X** |
| Protect livelihoods | - | X | X | - | - | X |
| Alternative livelihoods | X | - | X | - | - | X |
| Compensation | - | X | X | - | - | X |
| **Increase local acceptance** | **X** | **X** | **-** | **X** | **X** | **X** |
| Awareness | X | X | - | X | X | X |
| Education | - | X | - | X | X | - |
| Engagement | X | - | - | - | X | X |
| **Develop ecotourism** | **X** | **X** | **X** | **-** | **-** | **X** |

ITHCP project proposals addressed 53% of their corresponding NTAP action points on average. Categories that are the least addressed include the implementation of site-based management techniques (41%), investment in adequate project area facilities (54%), efforts to counteract poaching (53%), efforts to increase local acceptance towards tigers (51%), and the development of ecotourism (56%) (Figure 2). For half of the projects, their least addressed categories also scored the lowest in the CA|TS Lite assessment (Figure 3). However, low-scoring categories did not necessarily correspond with low CA|TS Lite scores. Categories that received insufficient CA|TS Lite scores usually coincided with an outdated NTAP or were poorly addressed in the corresponding NTAP or project proposal. In cases where the NTAP is outdated, the shortfalls could be due to the plan and not the project proposal itself, which still provides insight into what needs to be addressed. Note that Figure 3 includes a subset of the studied projects, because not all projects were assessed by CA|TS.

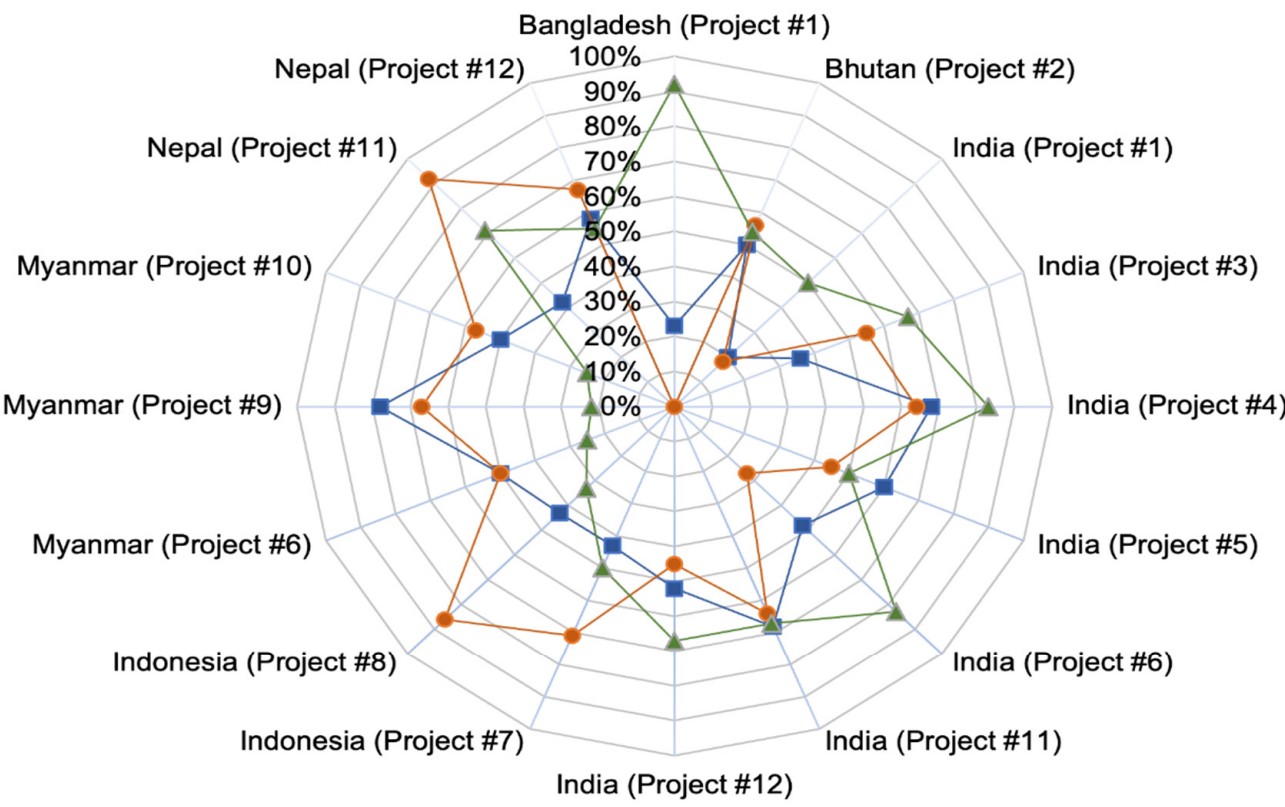

**Figure 2.** The extent to which ITHCP project proposals address the actions stated in their corresponding NTAPs that match with the ITHCP pillars.

Regions fluctuate in how much of each ITHCP pillar they address (Figure 4). In particular, efforts to engage with local communities are well addressed in some regions (Bangladesh, India, and Nepal), although not others (Bhutan, Indonesia, and Myanmar). There is also a wide range of CA|TS Lite scores among regions. Projects in India, Bhutan, and Nepal scored generally well (≥0.75), whereas projects in Bangladesh and Myanmar did not (≤0.5) (Figure 5). Numbers to support the radar graphs can be found in Supplementary Material Tables S8 and S9.

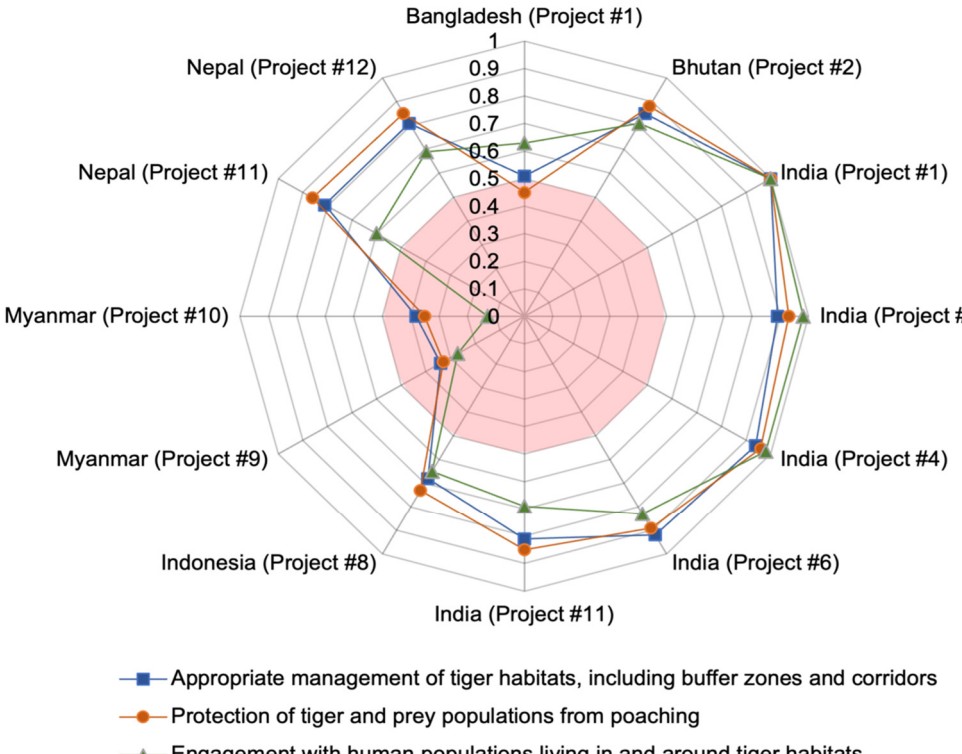

**Figure 3.** Mean CA | TS Lite assessment score for the ITHCP pillars and corresponding categories for each CA | TS-assessed ITHCP project: 1 = Recognized—Action Implemented, 0.75 = Recognized—Action Initiated, 0.5 = Recognized—Action in Planning Process, 0.25 = Recognized—No Action Initiated, 0 = Not Recognized—Action Absent. Categories that score below 0.5 (shown in light red) are considered to be insufficiently addressed [24].

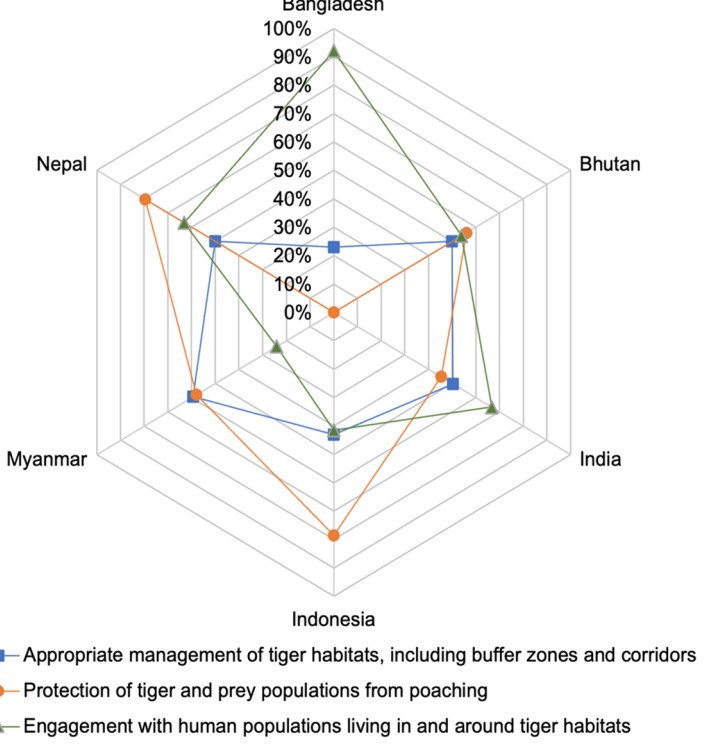

**Figure 4.** Average per country of the extent to which the proposals of their ITHCP project(s) address the actions stated in their corresponding NTAPs that match with the ITHCP pillars.

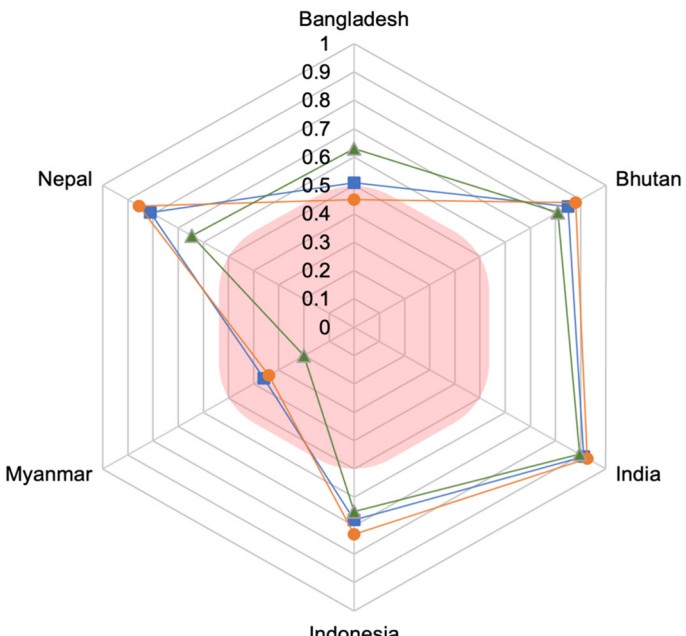

**Figure 5.** Average per country of mean CA|TS Lite assessment score for the ITHCP pillars: 1 = Recognized—Action Implemented, 0.75 = Recognized—Action Initiated, 0.5 = Recognized—Action in Planning Process, 0.25 = Recognized—No Action Initiated, 0 = Not Recognized—Action Absent. Categories that score below 0.5 (shown in light red) are considered to be insufficiently addressed [24].

With regard to tiger numbers, all project areas with tiger population data available showed an increase (Table 2). Bhutan shows the largest precentral increase (93%), and increases in other countries fluctuate heavily per project site. Data from Tiger Conservation Landscapes in Myanmar are missing. We acknowledge that we do not have all the metrics available to us in this study to do a full impact assessment with other measures such as density, or longer-term information, as that is beyond the scope of this study.

**Table 2.** Overview of change in tiger populations in each ITHCP project displayed in absolute numbers and percentages.

| Landscape/Country | Project | Baseline 2015 | Target | Final 2021 | Change (%) |
|---|---|---|---|---|---|
| India/Bangladesh | #1 | 182 | None | 202 | 11 |
| Bhutan | #2 | 12 | 18 | 23 | 93 |
| India | #3 | 15 | 16 | 20 | 33 |
| India | #4 | 190 | 240 | 310 | 63 |
| India | #5 | 12 | 20 | 17 | 42 |
| India/Myanmar | #6 | Unknown | None | 10 | - |
| Indonesia | #7 | 114 | 66 | 91 | 40 |
| Indonesia | #8 | 14 | 21 | 23 | 15 |
| Myanmar | #9 | Unknown | None | 8 | - |
| Myanmar | #10 | Unknown | None | 3 | - |
| India/Nepal | #11 | 89 | 107 | 169 | 90 |
| India/Nepal | #12 | 149 | None | 152 | 2 |

## 4. Discussion

Based on the ITHCP pillars, the ITHCP projects addressed around half of the NTAP action points on average (Supplementary Material Table S8). Categories that were the least addressed (<60%) include the implementation of management techniques (41%), investment into adequate facilities (54%), efforts to counteract poaching (53%), efforts to increase local acceptance (51%), and the development of ecotourism (56%) (Supplementary Material Table S8). In short, the projects never fully satisfied all the action points identified by the NTAPs. These discrepancies are mainly due to the following reasons:

- The ITHCP fulfills only a subset of the Global Tiger Recovery Programme, while most action plans refer to the entire Global Tiger Recovery Programme, inherently restricting the scope of ITHCP projects from the outset to fulfill NTAPs in their entirety. In other words, projects need to be self-contained to fulfil donor requirements and are, therefore, not a good reflection of broader action plans;
- Some project activities are restricted by external factors. Many actions concerning management techniques and law enforcement are heavily dependent on government policy and also operate within the restrictions placed by donor organizations [30]. Indeed, many of the actions that were poorly addressed are often partly or wholly the responsibility of government agencies, and not NGOs, and it is usually the NGO sector that applies for and implements ITHCP projects. This is why projects usually focus on the provision of infrastructure, training, and equipment to staff;
- Projects situated in multi-NGO landscapes may divide NTAP action points between them;
- Action points, such as investments into adequate facilities, may be too expensive to address within the scope of ITHCP projects;
- Projects are restricted by time, which makes it difficult to gain trust from the local communities to increase acceptance of conservation efforts [31];
- Ecotourism development is also dependent on external factors [32]. Not all countries have wildlife that is easily accessible, visible, or at a high density, reducing tiger-related ecotourism opportunities. Despite this, the extent to which projects develop sustainable ecotourism varies, suggesting that it is possible for projects to satisfy this category if more attention were paid to it. Since sustainable ecotourism benefits local communities as well as biodiversity, plans should be made to improve ecotourism where needed [33].

For example, one of the main discrepancies coming forward is the status of the transboundary Sundarbans project. On the Bangladesh side of the border, the project proposal only covered efforts to reduce human–wildlife conflict, improve local livelihoods, and increase local acceptance towards conservation activities (Figure 2). Consequently, implemented actions towards investment into staff and facilities, protection of tiger and prey populations, and ecotourism development were all deemed as insufficient by the CA|TS Lite assessment (Figure 3). A possible explanation for this is that, in Bangladesh, funding for conservation is limited, and activities need to be prioritized. It is globally recognized as a hotspot for human–tiger conflict and high poverty levels, and managing this conflict while working with local communities is a priority. Habitat restoration is often beyond the timeframe and financial scope of ITHCP projects, and localized restoration activities are not relevant here, again represented by the results where habitat-related measures are low scoring. Ecotourism is also in its infancy, with little investment to date [32]. Thus, there is a need for better, more sustainable, ecotourism development plans in Bangladesh to prevent harm to biodiversity and local communities [34].

In comparison, the Indian side of the Sundarbans was the highest assessed project by CA|TS Lite (Supplementary Material Table S9). However, as mentioned before, the Indian NTAP is stricter, with a longer history, and has been revised several times since the 1970s, highlighting the importance of effective action planning. It should also be noted that the Indian Government invests in tiger conservation at several orders of magnitude above that of many other Tiger Range Countries. Action plans in Bangladesh need further refinements, and conservation projects in the Bangladesh Sundarbans need to implement more actions

than those required by their NTAP to contribute more impact to the Global Tiger Recovery Programme. These transboundary differences show the importance of global action plans and a global framework to work against besides the national action plans.

Another notable gap identified was that two out of three projects in Myanmar were assessed to implement insufficient actions in almost every ITHCP category according to CA|TS Lite (Supplementary Material Table S9). The third project was not assessed by CA|TS Lite but it addressed the fewest national action points out of all three projects in Myanmar (Supplementary Material Table S8). One possible explanation is that the Myanmar NTAP was set to be implemented between 2003 and 2007 and is, therefore, extremely outdated. Fortunately, the government of Myanmar will release a new NTAP soon [35]. In addition, all the ministerial congress members hosted by Malaysia in February 2022 for Tiger Range Countries agreed to develop coherent national tiger action plans that are robust and aligned to a global, overarching, unifying framework. This framework is being developed through iterative dialogue between stakeholders such as donors, Tiger Range Country governments, and NGOs. This will improve the applicability and relevance of NTAPs to projects on the ground.

Of course, the best way to measure the progress of tiger habitat conservation is an increase in tiger populations [16]. However, many project areas currently do not have a clear baseline population to measure tiger numbers against. In addition, not all current estimates of tiger numbers are reliable, mainly because tigers range widely between transboundary sites [21]. According to the numbers we do have, most of the sites where ITHCP projects have been carried out have shown substantial increases in tiger populations from baselines beginning at the program's implementation period in 2015 to the end of 2021. However, it should be noted that these data are correlated only for the following reasons: firstly, tiger populations do not change rapidly enough for changes to be detected within the lifespan of many of these projects. Many of the population changes can be attributed to augmentation through dispersal events and source sink dynamics. The operational sites of many of the projects within the ITHCP program are often small in relation to the spatial scales of tiger movements. As a result, many projects operationally only cover a subset of sites within a park, and a fragment of a meaningful contiguous population. Projects within the ITHCP portfolio do not operate in isolation, but in a landscape where other donors, programs, and projects are in operation. Therefore, population changes cannot be attributed to single projects. However, the larger a project is, in terms of geographical and financial size, and the longer it is operational, the more likely it is to match more closely the broader objectives of the National Tiger Action Plan.

As a result, it is difficult to directly link project activities to needs within the context of individual landscapes and sites. This is an important finding from this cursory analysis. One solution to this problem would be to create resource need and resource input maps and compare the two. This would help provide an overview of Tiger Conservation Landscapes for project donors and government agencies. It is recommended that this is carried out to build on the analysis carried out here at programmatic levels.

CA|TS is one platform upon which resource needs, gaps and conservation activities can be centralized and evaluated on a site-by-site basis, especially since it aligns closely to the Global Tiger Recovery Programme [24]. Overall, the extent to which ITHCP projects addressed NTAPs is also reflected in the CA|TS Lite assessments of these projects. This suggests that CA|TS Lite effectively highlights gaps in project activities and can be useful as a management diagnostic tool. Additionally, the CA|TS -Log software is now available to tiger sites to further help with quantifying results and providing clear guidance for resource investment and site gap analysis [36]. Besides CA|TS Lite, other diagnostic tools can be used as well, such as the IUCN Green List, which is the first global standard for protected and conserved areas and offers locally relevant expert guidance to help achieve fair and effective nature conservation [37]. CA|TS and the Green List can be implemented together, as they complement each other and also fully align with the management effectiveness

tools MEETR, METT, or METT4 [38], which are widely used across the tiger sites in Tiger Range Countries.

However, for some projects, highly addressed NTAPs were still deemed to be insufficiently met by CA | TS Lite. This could be because proposed actions were not implemented, for, as mentioned before, having an activity in a proposal does not guarantee that it is carried out. Another explanation is that the actions required by the NTAP are not effective enough to satisfy the more stringent CA | TS Lite standards. In addition, some projects addressed NTAPs to some extent, but implemented actions were still deemed to be sufficient by CA | TS Lite. Possible explanations are that projects either implemented more actions than intended or NTAPs require more actions than necessary to establish good conservation practices. This is often the case in countries with a strong NTAP and a long history of tiger conservation, such as India [27]. Furthermore, some NTAPs were published after the project proposals were written. This was the case for the NTAPs of Bangladesh, Bhutan, and Nepal. Although both proposals and NTAPs ought to be drafted according to the tiger status and context of the country, this delay might explain some deviations between the proposals and NTAPs in these countries.

Conservation Assured standards, such as CA | TS, were originally established to provide robust metrics against which to measure the effectiveness of site-based protected area management. As a result, using them to assess the effectiveness of projects within those sites is one step removed and weakens this link. However, if appropriate metrics can be drawn out and linked directly to them, then to some extent such standards can be used to measure the effectiveness of projects within the context of broader site-based metrics. There are many other ways to improve such standards: firstly, they need to be scaled up to address broader landscape-level conservation needs to capture meta population conservation as opposed to focusing on individual isolated sites. Secondly, they need to address local community engagement and competing land tenure systems. Nonetheless, Conservation Assured standards have made a start in providing a common framework against which to implement conservation activities. Similar standards for other large predator species such as jaguars and river dolphins are also being rolled out. Addressing the problems faced by apex predators will contribute greatly to the conservation status of habitats and species across the globe. By conserving tiger habitats and associated protected and conserved areas, a whole range of other species could benefit through the "umbrella effect" [4].

The results from this study can also help inform plans to develop projects for other wide-ranging species besides predators that face similar threats, such as elephants. This will enable those projects to better address conservation concerns and ensure that resources are being effectively allocated. Our results can also inform project designers on how it is important to take regional differences into consideration when developing plans, and how it is beneficial to use multiple related tools to fill management gaps. This would not only improve the conservation of specific species, but that of broader habitats as well.

## 5. Conclusions

To conclude, we show that comparing project activities with national requirements is helpful for assessing resource needs and allocations to fill gaps in conservation management. Consequently, project design is a crucial step and should always be based on global and national action plans and best practice standards, and in the context of the activities of other projects in the landscape. Since the NTAPs form the foundation of the Global Tiger Recovery Programme, it is important that all NTAPs are followed by projects to increase wild tiger numbers. Given the geological differences across landscapes, projects have different needs. If standards such as conservation assured can adapt to address these needs they can be a crucial tool that helps to align global plans, national plans and projects, especially as making global and national plans is an evidence based process. Therefore, devoting time and resources towards good conservation standards is profitable for both governments and donors, as the latter are also looking for allocation of resources to fill conservation gaps.

Furthermore, when it is known to what extent the intentions of projects match with the national plans, the next step is to see to what extent these intentions have been realized through implementation. Furthermore, research must be conducted to see whether the identified gaps exist in reality or have been addressed some other way through non-project means; for example, through government programs. Where action points have truly been unaddressed, further investigation is needed to ascertain whether or not they do need to be addressed within the context of the threats faced by a particular threatened species in that particular setting.

Depending on the needs across the different sites and levels of management, a combination of tools and good practice guidelines can be adopted to achieve desired conservation outputs and outcomes.

**Supplementary Materials:** The following supporting information can be downloaded at: https://www.mdpi.com/article/10.3390/land11122326/s1, Table S1: Analysis of how many NTAP action points the project proposal of the ITHCP project in Bangladesh addresses; Table S2: Analysis of how many NTAP action points the project proposal of the ITHCP project in Bhutan addresses; Table S3: Analysis of how many NTAP action points the project proposal of the ITHCP projects in India address; Table S4: Analysis of how many NTAP action points the project proposal of the ITHCP projects in Indonesia address; Table S5: Analysis of how many NTAP action points the project proposal of the ITHCP projects in Myanmar address; Table S6: Analysis of how many NTAP action points the project proposal of the ITHCP projects in Nepal address; Table S7: CA | TS Light standards used to determine mean CA | TS accreditation score per category of the three ITHCP pillars [24]; Table S8: Overview of how ITHCP projects addressed their respective NTAPs; Table S9: Overview of how ITHCP projects scored on the ITHCP categories according to CA | TS Lite.

**Author Contributions:** Conceptualization, J.D. and S.R.; methodology, J.D. and S.R.; formal analysis, J.D., M.S., M.K.S.P. and S.R.; data curation, J.D., M.S. and S.R.; writing—original draft preparation, J.D. and S.R.; writing—review and editing J.D., M.S., M.K.S.P. and S.R.; visualization, J.D., M.S., M.K.S.P. and S.R.; supervision, M.K.S.P. and S.R. All authors have read and agreed to the published version of the manuscript.

**Funding:** No external funding was received to carry out this study.

**Institutional Review Board Statement:** Not applicable.

**Informed Consent Statement:** Not applicable.

**Data Availability Statement:** Not applicable.

**Acknowledgments:** The authors would like to thank the recipients of ITHCP funding for their efforts in developing and implementing large and complex projects under difficult circumstances. Additionally, the authors want to thank all the Tiger Range Governments and the CA | TS Partnership. The authors would also like to thank the ITHCP Secretariat and staff of IUCN in HQ and country offices throughout Asia involved in the program. Lastly, the authors would like to acknowledge the countries, agencies, site managers, and teams involved in the CA | TS process.

**Conflicts of Interest:** The authors declare no conflict of interest. This research was supported by the ITHCP which is funded by the German development bank Kfw. The authors declare that they have no known competing financial interests or personal relationships which have or could be perceived to have influenced the work reported in this article.

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
