# Peer review of "Assessing Project Proposals Based on National and Global Tiger Action Plans: Lessons from the Integrated Tiger Habitat Conservation Programme (ITHCP)"

_land, doi:10.3390/land11122326_

Round 1
Reviewer 1 Report
The authors strive to evaluate the likely effectiveness of tiger projects by comparing the funded proposals against international and national standard of needs for tiger conservation. I really like the emphases on evaluation. I think that it is adequate to do what the authors did and evaluate when plans do not hit all of the core challenges. However, the authors do not seem to link the plans to a process or a governance structure or a conceptual model used to make the proposal. Including these would make this paper so much richer by being able to assess why projects from different entities (e.g., NGO’s) are more or less able to get the job done. I strongly suggest that the authors look at the Conservation Standards (CS) (Conservation Measures Partnership, ver 4.0 2022 (I think)). These CS use conceptual models to ink actions to threats through a Theory of Change. This would give the authors to the power to ask not whether or not the proposals ticked all the boxes, but whether or not the proposal was sensible given the situation. That would be a better paper.
Line 110. New ‘method’? Maybe new process, new endeavor? Effectiveness evaluation is not a new method. It isn’t done much and it might be a new endeavor on tigers, but “new method” just isn’t accurate. You may have used a new method for effectiveness evaluation. But, there, you haven’t told us that yet, so it is confusing. (after reading the whole thing, I implore you to drop the use of the word ‘new’ or ‘novel’. This method is fine, but neither new nor novel. It is standard evaluation.
Line 114. Already my head is reeling from an abundance of acronyms. Please try to find ones that you use less than ~5 times and then spell those out. Focus on using the ones that MOST help you tell the story. As it is, it is too much to keep track of.
Line 120.if you are evaluating proposals, then you are evaluating something like the presumed effectiveness, and not the actual effectiveness.
Line 122. Evaluate them for … what? Say it in this sentence.
Line 145. What about partially addressed? It seems likely that there would be a lot of that. Yet, no score for it?
Line 174. You describe a scoring process, then here in Table 1 we get X’s instead of 1’s and 0’s. Why?
Line 230. I found these results to be disappointing. Yes, it is descriptive. And, you are telling us (and these countries) which are developing plans that conform to their NAPT’s and the ITHCP…. But, we could learn so much more that could be helpful. Specifically, can you predict which plans are more or less comprehensive based on a set of predictors. Potential process predictors: who was engaged in writing the proposal and how it was done (govt only, NGO only, govt + NGO, with / w/o stakeholder input etc). Potential Situational predictors: where tigers are more or less under duress currently. I am certain that you can think of more. But, this would make this a generally interesting paper rather than a paper that is solely interesting to those who wrote the tiger plans.
Line 240-250. A priori, I would expect that not all problems and problems everywhere, and hence not all plans should contain all aspects. I would expect that each proposal would address all issues and report that they are focusing resources on the problems that vex the region. Thus, I would recommend awarding a score of 1.0 for plans where they develop a conceptual model and have a good rationale for NOT addressing some stress to tigers, and then not addressing it. Thus, I think the evaluation should not treat all proposals as if they are equal. I think you do that, to some extent, by linking action proposal to national priority plans. But, I think you need to be a LOT more explicit about this, and perhaps likely need to rescore proposals.
Line 261. Here is a perfect example of a justifiable reason to not address an issue. So, what would you recommend? That they plan for things that they can’t do, or engage partners so that they can do what needs to be done. I think that you could benefit by examining the Conservation Standards as a model of linking actions to outcomes. There are models for this. Theories of change. So, then you would be evaluating whether they have a conceptual model, a theory of change and ask if the plan is consistent with the conceptual model.
Line 278. Yes, well, then what are you trying to do? So, This paper would benefit with a clearer frame for the objective of your work. Is it really to assess conservation gaps? If so, then Fig 2 should go away and Fig 3-6 should be clarified and specified. These are showing what projects are lacking some key actions,. Then, might you go back and try and evaluate why?
Line 290-300. And there you go. If you simply count the deficiencies without linking them to a conceptual model, you end up with a litany of vague reasons why or how they could be better.
This discussion is ~3 times too long… So are the conclusions.
Reviewer 2 Report
This study has merit and deserves publication. It addresses a flagship and transnational conservation initiative – the global tiger population. It reviews conservation actions in eight key countries but is not judgemental. Rather it seeks to derive a metric to facilitate judgements of donors and policy makers in their initiatives to conserve and augment tiger populations. To this end it is largely successful. The introduction sets out the rationale for the study well. The methods are clear, and the results presented elegantly in figures and tables. However, the discussion can be improved. It commences with clarity and conciseness and then becomes ponderous with very long convoluted sentences. The important issues under consideration - mainly, historic and contemporary factors leading to variance and some failure to achieve conservation goals – are obfuscated. Thus, I suggest this section be reviewed and at the least the sentence structure made more concise. Otherwise, I noted a few minor errors as follows:
Line 71: not clear what a ‘universal log frame’ is – some database?
Line 78-79: no action for (iii) – (i) management, (ii) protection, (iii) ?
Line 154: indicators
Line 273: inconsistency in text with eco-tourism and ecotourism
Line 326: more impact to
Line 345: beginning at
Line 365: too short
Line 369: larger a project
Line 399: globally
Round 2
Reviewer 1 Report
After reading your responses to my review, I would like to say two things. First, I apologize. The tone came across as a bit grumpy. I put a lot of work into that review because I think that this is really important work. I still think that there is further work that you could do that uses a theory of change approach to map proposed actions through the logic model for why the countries think that these are the actions that will meet their priority challenges. But, perhaps that is for a different paper.
Second, yes, I think I mis-interepreted the aims of this paper, which are more circumscribed than I perhaps was thinking.
I think that you have done a much better job of describing your work in this version
Author Response
Dear Reviewer 1,
We thank you kindly for your fast and positive response to our revised version of this manuscript. We did not receive your initial feedback as grumpy. However, we are incredibly pleased to hear that the manuscript now seems to be to your satisfaction.
Kind regards and once again thank you for your helpful feedback.